# Effect of THz Waves of Different Orientations on K^+^ Permeation Efficiency in the KcsA Channel

**DOI:** 10.3390/ijms25010429

**Published:** 2023-12-28

**Authors:** Yize Wang, Hongguang Wang, Wen Ding, Xiaofei Zhao, Yongdong Li, Chunliang Liu

**Affiliations:** 1Key Laboratory for Physical Electronics and Devices of the Ministry of Education, Xi’an Jiaotong University, Xi’an 710049, China; wangyize@stu.xjtu.edu.cn (Y.W.); dingwen@mail.xjtu.edu.cn (W.D.); zhaoxiaofei@stu.xjtu.edu.cn (X.Z.); leyond@mail.xjtu.edu.cn (Y.L.); chlliu@mail.xjtu.edu.cn (C.L.); 2School of Electronic Science and Engineering, Xi’an Jiaotong University, Xi’an 710049, China

**Keywords:** KcsA channel, ion flux, dihedral angle, EM wave

## Abstract

Potassium (K) channels show the highest variability and most frequent alterations in expression in many tumor types, and modulation of K^+^ channels may represent a new window for cancer therapy. In previous work, we found that a terahertz (THz) field incident along the *z*-axis with a frequency of 51.87 THz increased the ion flux through K^+^ channels. In practice, it is difficult to ensure that the incident electromagnetic (EM) wave is strictly parallel to the direction of channel ion flow. In this paper, we found by changing the direction of the applied electric field that the EM wave of a specific frequency has the largest ion flux when the incident direction is along the ion flow, and the smallest ion flux when the incident direction is perpendicular to the ion flow, and that overall the EM wave of this frequency enhances the ion flow of the K^+^ channel. Changes in the direction of the applied field at a specific frequency affect the stability of the φ dihedral angle of the GLY77 residue and alter the ion permeation mechanism in the selectivity filter (SF) region, thus affecting the ion flux. Therefore, this frequency can be used to modulate K^+^ fluxes by THz waves to cause rapid apoptosis in potassium-overloaded tumor cells. This approach consequently represents an important tool for the treatment of cancer and is expected to be applied in practical therapy.

## 1. Introduction

Ion channels are key regulators of cellular homeostasis in excitable and non-excitable cells and regulate important physiological processes such as electrical signaling, gene expression, and cellular signaling pathways [1,2]. The preponderance of evidence linking aberrant ion channel activity to carcinogenesis, migration and invasion of cancer has led to the perception of cancer as a channelopathy [3,4,5]. Potassium channels are involved in cell proliferation, cell cycle transition and apoptosis, and exhibit the highest variability and most frequent expression changes among many tumor types [6,7,8,9]. Different experimental approaches have demonstrated the relationship between potassium channel blockade and anticancer effects [10,11]. Therefore, modulation of potassium channel function has been recognized as a potential approach for the treatment of different cancers.

Potassium channels have strong selectivity for K^+^ and an osmotic efficiency for K^+^ close to the diffusion limit [1,12,13,14,15]. Common to all potassium channels is the highly conserved SF region, a portion of the residue sequence is T-V-G-Y-G. The SF region underlies the precise selection of K^+^ and the high conductance of potassium channels [12,16,17]. The high-resolution structure of KcsA, a prokaryotic potassium channel, provides a direct structural basis for probing the function and behavior of eukaryotic potassium channels. The overall structural architecture of KcsA is a cross-symmetric tetramer with four subunits surrounding a central channel [18,19,20]. The channel can be divided into three regions from intracellular to extracellular: gate, Cavity, and the SF. The crystal structure of the potassium channel derived by MacKinnon and colleagues shows a string of distinct ion-binding sites within the cylindrical SF, as shown in Figure 1a [12,18]. Instead of forming a continuous chain, they suggest that the potassium ions are alternately bound to intervening water molecules, which weaken the large Coulomb repulsive forces generated by the direct contact of the cations, as shown in Figure 1b [14,21,22,23]. An alternative view is presented by Köpfer et al. In their extensive molecular dynamics simulations of K^+^ conduction through various potassium channels, they found that K^+^ ions contact each other directly along the single file, with few water molecules appearing in between, as shown in Figure 1c [23,24,25]. They suggested that direct Coulomb knock-on between the contacting cations enhanced ionic conduction. Sumikama et al. proposed a new view, queueing arrival and release mechanism. They argued that the mechanism that occurs during the transport of K^+^ is related to its concentration in the system, that the higher the concentration of K^+^ the fewer water molecules there will be at the SF site, and that there should not be a single permeation mechanism [26]. In our previous work, we found that the direct Coulomb knock-on mechanism is the most efficient for the transport rate of K^+^ and is the dominant permeation mechanism throughout the simulation [27].

Most of the low-frequency biological motions are in the same range as THz and infrared signals [28,29]. The frequency region from 0.5 to 100 THz is defined as the broad-sense THz EM wave, and is an effective way to modulate the physicochemical and dynamic behavior of biological macromolecules [30]. Olshevskaya and colleagues irradiated phloem nerve cells in static water with THz waves using a free-electron laser and showed that THz waves induced a change in membrane potential with a power density of 0.3 mW/cm^2^ [31]. THz waves significantly interfered with the naturally occurring localized strand segregation of the double-helix DNA, which could alter functions such as gene expression of the DNA [32]. Stimulation of calcium channels by 42.55 THz EM waves can significantly enhance the selectivity and conductivity of calcium ions, thereby correcting the calcium current reduction caused by calcium channel defects [33]. In our previous work, a THz field with a frequency of 51.87 THz incident along the direction of the ion flow can significantly the K^+^ flow in the KcsA channel, thus inducing rapid apoptosis of tumor cells with potassium overload [27].

In the experiments of Xi Liu et al., K^+^ current was increased by the applied THz waves with a wavelength λ of 5.6 μm to the recording cells [34]. In our previous work, we can see that when the frequency of the applied electric field is 51.87 THz, the ion flux of the potassium channel is significantly enhanced, and in addition, the incident direction of the applied electric field is along the direction of the ion current. In practice, it is difficult for us to just add THz waves in a particular direction of a channel [27]. In this paper, we verify the effects of 51.87 THz electromagnetic waves in different directions on the ion flux of the KcsA channel and further explore the mechanism of THz waves on the KcsA channel. The means of THz modulation of KcsA channel current will be more precise and efficient.

## 2. Results

In our previous work, we found that when the amplitude of the applied electric field was at 0.4 V/nm and the frequency was 51.87 THz, the K^+^ flux in the KcsA channel was 1.8 times higher than that without the applied electric field, and the ion current of KcsA was significantly improved. The frequency of 51.87 THz is the strongest absorption peak in the infrared absorption spectrum of carbonyl oxide in the SF region (Appendix A). THz electromagnetic waves are resonantly absorbed by aqueous solutions or proteins, and the frequency of 51.87 THz avoids the strong absorption spectral range of bulk water, resulting in a limited thermal effect from bulk water [27]. Meanwhile, previous work only considered the electric field direction along the *z*-axis, i.e., the direction of the channel current. Therefore, in this paper, we further investigate the intrinsic mechanism by which the applied electric field affects the ion channel by comparing the change in channel characteristics when the same frequency and amplitude, but different directions of the applied electric field, are used.

### 2.1. Ion Flux in the KcsA Channel

We chose to simulate the applied electric field with an amplitude of 0.4 V/nm, a frequency of 51.87 THz, and an incidence direction of the electric field of θ, which is the angle between the incident electric field and the *z*-axis (in the XOZ plane), as 0°, 30°, 45°, 60°, and 90°, respectively. We simulated the case of the channel with the applied electric field at different angles independently several times, with a total simulation time of 3.2 us for each angle, and then statistically averaged the simulation results. Figure 2 shows the relationship between the ion flux and current and the angle θ of the applied electric field when the amplitude of the applied electric field is 0.4 V/nm and the frequency is 51.87 THz, where E = 0 is the case without the applied field. It can be seen that when the electric field is incident along the *z*-axis, i.e., along the flow direction of K^+^, the ion flux is the largest, which is about 1.8 times that in the absence of the applied field. When the electric field is incident along the *x*-axis, i.e., perpendicular to the flow direction of K^+^, the ion flux is the smallest, which is slightly smaller than that without the applied field. Therefore, a change in the direction of an applied electric field of the same frequency and amplitude also affects the K^+^ flux in the KcsA channel. In experimental applications, the beams of THz lasers are generally at the millimeter scale, and the sizes of eukaryotic and prokaryotic cells are at the micrometer scale, which makes it difficult to ensure that the direction of the THz wave incident on each K^+^ channel is maintained [34]. From examining the relationship between the direction of the applied electric field and the ion current in Figure 2b, it can be seen that even though it is difficult to ensure that the direction of the applied electric field is maintained strictly along the direction of the ion current, the electric field at this frequency still increases the K^+^ current in the whole cell.

### 2.2. Density Distribution of K^+^ at SF Sites

The selectivity filter region of the KcsA channel is critical for efficient selection and transport of K^+^ by the channel, and we further analyzed the charge density distribution of K^+^ ions in the SF region under the action of electric fields of specific frequencies in different directions, as shown in Figure 3. K^+^ ions flow into the SF region from the cavity region, firstly reaching the S_4_ site, and then passing through the “S_3_–S_2_–S_1_–S_0_” sites in turn. The number of K^+^ ions conducted by the KcsA channel over a period of 3.2 μs varied considerably under the action of an applied electric field in different directions. When θ = 0°, the maximum number of K^+^ ions conducted by the channel was 587; when θ = 90°, the minimum number of K^+^ ions conducted by the channel was 374. From examining the K^+^ density distribution in Figure 3b, it can be seen that the applied electric field at different angles has little effect on the K^+^ distribution in S_4_ and S_3_. Starting from the S_2_ site, the ion distribution changes, with the highest K^+^ density in the absence of the applied electric field and the lowest K^+^ density at θ = 90°. At the S_1_ and S_0_ sites, the distribution n of K^+^ in the SF region appears to change even more, with the lowest density of K^+^ at the S_1_ site in the case of no applied field, followed by the case of θ = 90°, while the lowest density of K^+^ at the S_0_ site is simulated at θ = 90°.

### 2.3. Two Exotic States Slow down the Rate of Ion Permeation

We further analyzed the state of K^+^ ions at the SF sites as they pass through the channel. Based on previous articles, it can be seen that the high-speed transport of K^+^ via potassium channels mainly passes via the mechanism of direct Coulomb knock-on [24,25,27], as shown in Figure 4a. K^+^ is dehydrated from the Cavity region into the SF region and then passes through each site sequentially to complete single permeation. When the channel structure is unstable, some other permeation mechanism occurs, such as between neighboring K^+^ in the SF being spaced apart by water molecules. We analyzed the ion configuration states in the SF region under the action of an applied electric field with different orientations but the same frequency (51.87 THz) by counting the simulation results from every 5 ns. It is found that two singular states tend to appear during the simulation at θ = 90°, as shown in Figure 3b. K^+^ always stays in the SF region in a state occupying the ‘S_4_, S_3_’ or ‘S_3_, S_2_’ sites, and experiences difficulty in reaching the next site to rapidly complete the permeation of K^+^. The probability of the two states ‘S_4_, S_3_’, ‘S_3_, S_2_’ appearing in the simulation with θ = 90° is about 31% and the least number of ion permeation events within 3.2 μs at this angle. In contrast, the probability of these two states appearing in the simulation with the electric field at other angles as well as in the simulation without the applied field is lower than 15%, and the ion permeation events are both increased compared to the case without the applied electric field.

For the cases of applied electric field direction θ = 0° and θ = 90°, we randomly selected the simulation results for a period of time to analyze the ion trajectories, as shown in Figure 5. At θ = 0°, K^+^ in the channel mainly passes through the SF region in the mode of direct Coulomb knock-on, and the channel realizes efficient transport of K^+^. At θ = 90°, K^+^ also mainly passes through the SF region in the mechanism of direct knock-on to realize ion permeation, but the appearance of (S_3_, S_2_) and (S_4_, S_3_) states slows down the rate of K^+^ through the channel. As can be seen from Figure 5, when potassium ions appear in the SF in these two states, K^+^ takes a longer time to reach the next site and stay in the S_2_ or S_3_ site for a long time, while K^+^ passes through the channel rapidly after reaching the next site.

### 2.4. The Dihedral Angle of the GLY77 Residue

By observing the permeation state of K^+^ at the SF site in VMD [35] (Appendix A), we found that when K^+^ stays at the S_2_ or S_3_ site for a long time and it is difficult to reach the next site, it is mainly due to the reversal of the carbonyl oxygen atom at the next site, and K^+^ is unable to coordinate with the eight oxygen atoms. Therefore, we analyzed the dihedral angles of several sites in the SF region to explore the effect of the electric field direction on the channel structure [36,37]. The SF region of the KcsA channel is formed by four channel subunits at the interface, which is a four-fold symmetric structure, thus there will be four symmetrical residues and their dihedral angles, which are only shown on one subunit in this paper (Appendix A). Figure 6 shows the distribution of dihedral angles φ for different residues on one subunit with the applied electric fields of the amplitude 0.4 V/nm and the frequency 51.87 THz but different angles of incidence. It can be seen that the changes in the angles of incidence of the applied electric fields have no effect on the distribution of dihedral angles at the S_4_ and S_3_ sites, consisting of THR75 and VAL76, where the K^+^ arrives first. Starting from residue GLY77, the dihedral angles were distorted to different degrees by different angles of applied electric fields, and the dihedral angles of GLY77 were difficult to maintain in a limited range of values. In the cases where no electric field is applied (black solid line) and with the applied electric field of θ = 90° (dark blue dash-dotted line), the dihedral angle of GLY77 is difficult to maintain within a finite range of values and cannot be stabilized within the desired angle range. In the corresponding ion flux, the ion flux is relatively small for both the cases without the applied field and the case with θ = 90°. The dihedral angle of GLY77 is most stabilized at the case of the θ = 0° (red dashed line), which corresponds to the highest ion flux. The TYR78 and GLY79 residues under the chain reaction were also distorted to varying degrees when the case without the applied electric field and the case with the applied field of θ = 90°, leading to fluctuations in the channel structure, which in turn affected the permeation of K^+^. The GLY77 residue was able to be a disruptor of the α-helical conformation because its side-chain group was too small, making the dihedral angle take on too large a range of values. Therefore, the applied electric field affects the distribution of its dihedral angle, which in turn affects the ion flux through the channel.

## 3. Discussion

In a previous study, we found that after applying the THz electric field with a frequency of 51.87 THz to the KcsA channel, the transport rate of K^+^ was significantly increased, measuring 1.8 times higher than the case without the applied electric field. The direction of the applied electric field was along the direction of K^+^ flow. In experiments where THz lasers are incident on cells, it is difficult to ensure that the direction of the electric field in the electromagnetic wave is strictly parallel to the direction of ion flow. In this paper, by changing the direction of the electric field at a specific frequency, it was found that the electromagnetic wave of 51.87 THz still increased the K^+^ flux in cells when the propagation direction of the THz wave could not be determined.

The change in the direction of the electric field simultaneously alters the transport mechanism within the SF, with the faster-transporting direct knock-on mechanism having the highest probability of occurring at θ = 0° and the corresponding ion flux being the largest. The slower-transporting soft knock-on mechanism has the highest probability of appearing at θ = 90°, and the corresponding ion flux is the smallest. The intrinsic mechanism is that the change in the direction of the applied field affects the dihedral angle of GLY77, which shows a large aberration and is not stabilized within a reasonable angle in the case of θ = 90°. The distortion of GLY77 leads to instability of the channel structure, which makes it difficult for potassium ions to reach the next site quickly, thus slowing down the permeation rate of K^+^.

This finding improves the efficiency of ion channel regulation by THz waves. A large body of evidence shows that abnormal ion channel activity is associated with carcinogenesis, cancer migration, and invasion, which presents THz wave interference with ion channel modulation as a new direction for cancer therapy. Existing THz lasers employ millimeter-sized beams, while tumor cells tend to be larger than normal cells but also smaller than millimeters [34]. Thus, THz lasers can enable precise localized irradiation of tumors (except for small cell carcinoma of the lung, etc.). Tumor cells have a larger water content than normal cells, and the same THz field action will have a stronger effect on the ion channels due to polarization, thus reducing the effect on normal cells [38,39]. In the future, THz may become a promising alternative to adjuvant therapy or traditional anti-cancer methods. Our results provide a basis for the technical feasibility and design of subsequent devices capable of delivering THz fields in biological environments at the desired direction and frequency.

## 4. Materials and Methods

There are two main methods for constructing transmembrane potentials in molecular simulations: the constant electric field method (CEF) and the ion imbalance method (IIMB) [40,41,42,43,44]. The CEF method simulates the force of the transmembrane voltage by applying an electric field force to each charged particle. In the IIMB method, the voltage is generated by a small charge difference ∆q across the membrane, which is closer to modeling the environment in a real biological system than the CEF method (Appendix A). In this paper, we will use a combined CEF-IIMB model to explore the effect of a THz electric field at a specific frequency on the ion transport in potassium channels. The IIMB method is used to construct the membrane voltage that drives the ion flux, and the CEF method is used to apply an external THz electric field to the system (Appendix A).

We selected an open KcsA channel from Protein Data Bank with ID 5VK6 [20]. The channel was embedded in a patch of palmitoyl-oleoyl-phosphatidylcholine (POPC) membrane containing 134 lipid molecules, which were then hydrated by 10,359 TIP3P water molecules, and KCl (0.3 M) was also added. This resulted in an 8 × 8 × 9.4 nm^3^ simulation system. This system was set up by the CHARMM-GUI web server [45]. MD simulations were performed using GROMACS 2019.4 with CHARMM36 force field. Periodic boundary conditions were applied in the XYZ direction [46]. PME was used to treat electrostatic interactions exceeding the 1 nm cutoff and set the cutoff of vdW interactions to 1 nm. The LINCS constraint algorithm was used to reset bonds after an unconstrained update of 2 fs. The pressure and temperature were held at 1 atm and 300 K by the semi-isotropic Parrinello–Rahman barostat and the v-rescale thermostat, respectively. The systems performed a steepest descent energy minimization and then equilibrated until the system stabilized. In the 5 ns of the NVT and 15 ns of NPT equilibration simulations, the heavy atoms of proteins were restrained with a force constant of 1000 kJ mol^−1^ nm^−2^ to their starting positions. Lipids, ions, and water were allowed to move freely during equilibration (Appendix A).

## 5. Conclusions

We use a combined CEF-IIMB model to simulate the application of THz waves of a specific frequency to the KcsA channel, which conducts K^+^ at a high rate. In previous work, it was found that a THz field with a specific frequency of 51.8 THz incident along the *z*-axis increased ion flux through the potassium channel. In practice, it is difficult to ensure that the incident EM wave is strictly parallel to the direction of ion flow through the channel. In this paper, we found by varying the direction of the applied electric field that the ion flux in the channel is maximum when θ = 0° and minimum when θ = 90°, but the overall ion flow through the potassium channel is increased. The change in the direction of the electric field at a specific frequency affects the alteration of the channel structure, which is fundamentally due to the difficulty in stabilizing the φ dihedral angle of the GLY77 residue within a reasonable range in simulations with θ = 90°. This leads to an interlocking reversal of the S_2_ as well as S_1_ sites, which makes it difficult for K^+^ to pass through rapidly, and the ion flux in the channel decreases.

The findings in this paper can serve as one of the pieces of evidence supporting the great potential of K^+^ channels as targets for personalized cancer therapy [3,7]. Electromagnetic waves of specific frequencies can accelerate the apoptosis of potassium-loaded tumor cells by modulating ion flow through ion channels. Due to the complexity of the simulation system, we only performed simulations of ion permeation through a single channel. In the future, designing experiments with multiple K^+^ channel simulations or simultaneous presence of K^+^, Na^+^, and Ca^2+^ channels is necessary, and we should at the same time improve the computational time to the order of microseconds or even higher.

## Figures and Tables

**Figure 1 ijms-25-00429-f001:**
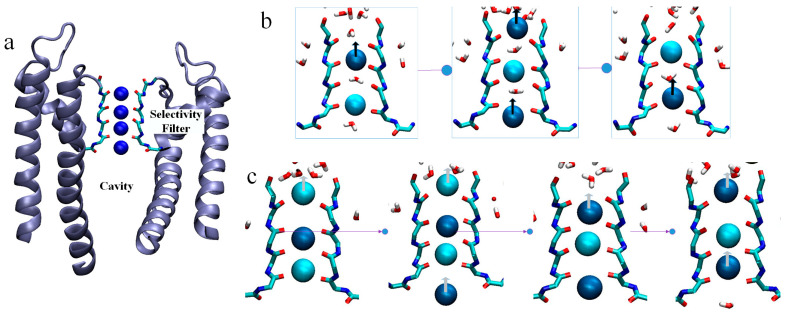
KcsA channel. (**a**) A cut through the KcsA channel shows the ion-binding sites (blue). (**b**) Soft knock-on mechanism. Shades of blue balls represent neighboring K^+^. The red–white combinations represent water molecules. The black arrow represents the direction of flow of K^+^ by a soft knock-on mechanism. (**c**) Direct knock-on mechanism. The white arrow represents the direction of flow of K^+^ by a direct knock-on mechanism.

**Figure 2 ijms-25-00429-f002:**
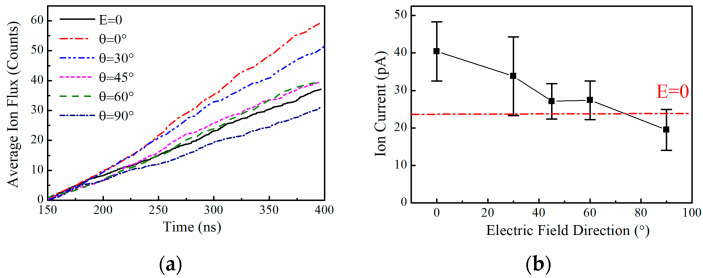
Effect of THz electric fields of different orientations at specific frequencies on ion flux or current in the KcsA channel. (**a**) K^+^ flux in the KcsA channel varies with the angle θ of the electric field when the applied electric field frequency is 51.87 THz and the amplitude is 0.4 V/nm. (**b**) K^+^ current of the KcsA channel varies with the angle of the electric field when the applied electric field frequency is 51.87 THz and the amplitude is 0.4 V/nm. The short red-dotted line is the K^+^ of the channel without the applied field current, which is about 23.6 pA.

**Figure 3 ijms-25-00429-f003:**
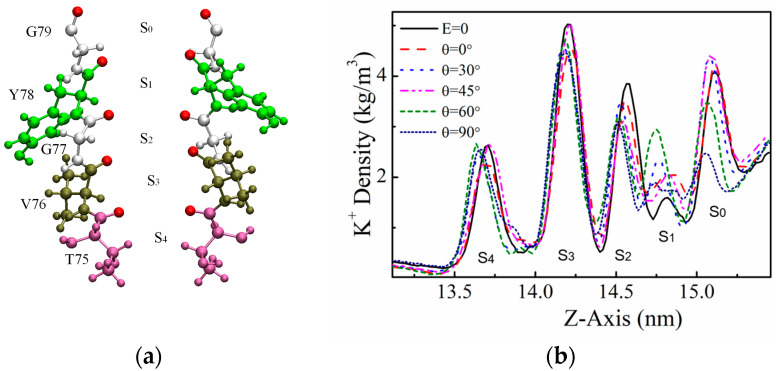
K^+^ density distribution within the SF region in the KcsA channel. (**a**) The SF region of the KcsA channel, where each different color represents a different residue, and each site is composed of oxygen atoms contributed by the -C=O group of every two neighboring residues. (**b**) Density distributions of K^+^ at each site within the SF under the action of an applied electric field of different orientations, and at a specific frequency. The solid black line represents the result without an applied electric field.

**Figure 4 ijms-25-00429-f004:**
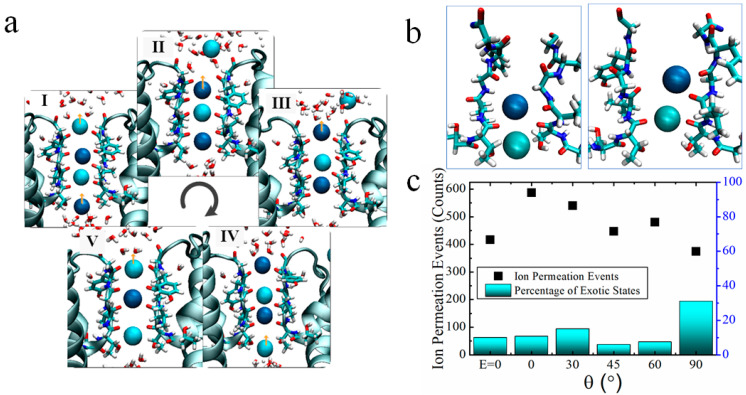
Ion permeation process. (**a**) The permeation of K^+^ in KcsA channel is mainly based on the direct knock-on mechanism. I–V represents the cyclic process of K^+^ passing through the channel. Where neighboring K^+^ are distinguished by different shades of blue balls. The red-white combinations are water molecules. The arrows represent the direction of K^+^ flow. K^+^ dehydrates from the cavity region into the SF region to reach the S_4_ site, and then passes through each site in turn, thus completing an ion permeation. (**b**) In simulations with the electric field direction of θ = 90°, in addition to direct Coulomb knock-on, two states appear more frequently, with K^+^ occupying the (S_3_, S_2_), or (S_4_, S_3_) sites, respectively (both blue balls are K^+^). (**c**) Probability distributions (bars) of the two exotic states in simulations with different electric field orientations. Distribution of the number of ion permeation events in simulations with different electric field orientations (dotted). The horizontal coordinates are the different angles of incidence of the applied electric fields, and E = 0 represents the case without the applied electric field.

**Figure 5 ijms-25-00429-f005:**
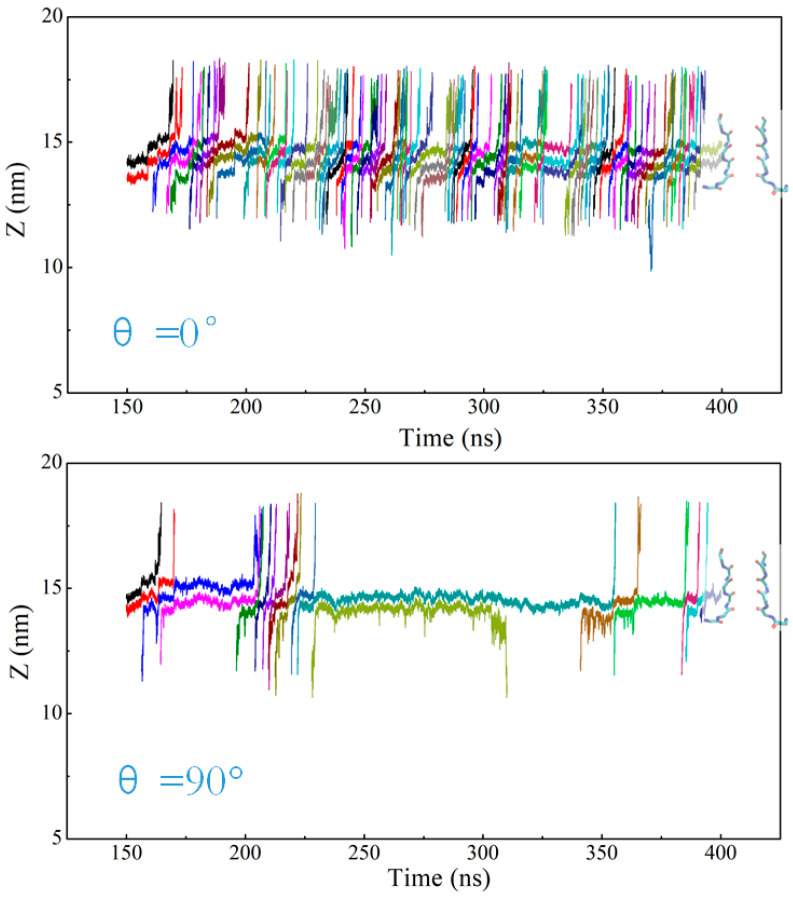
Trajectory of K^+^ in the SF. When the angle of the applied electric field is θ = 0° or θ = 90°, the trajectories in SF are randomly selected for a period of 150–400 ns simulation time, and the lines of different colors represent the trajectories of different K^+^ ions.

**Figure 6 ijms-25-00429-f006:**
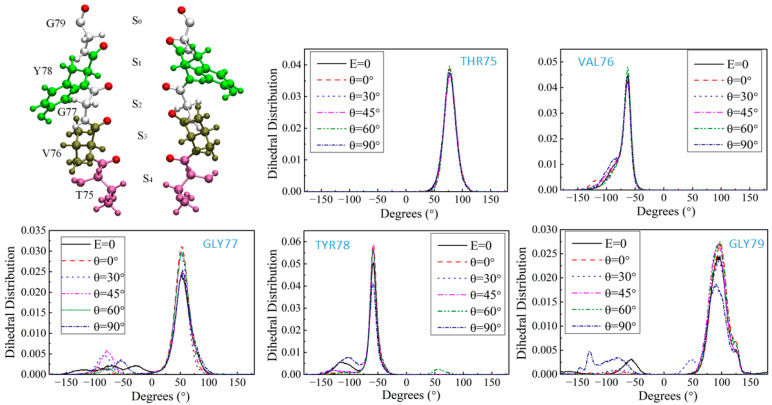
φ dihedral angle distribution for different residues. Different colors represent different residues in the SF region. The black solid line shows the dihedral angle distribution without the applied electric field; the dark blue dash-dot line shows the dihedral angle distribution with the applied field of θ = 90°.

## Data Availability

The data presented in this study are available in article and Appendix A.

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
