# Peer review of "Effect of THz Waves of Different Orientations on K+ Permeation Efficiency in the KcsA Channel"

_ijms, 2023, doi:10.3390/ijms25010429_

Round 1

Reviewer 1 Report

Comments and Suggestions for Authors

As a reviewer, I find the manuscript's exploration of the effects of terahertz (THz) electric fields on potassium (K+) flux through KcsA channels to be a novel and significant contribution to the field of ion channel biophysics and its potential application in cancer therapy. The combined CEF-IIMB simulation approach is well-justified and represents a robust methodology. However, the manuscript would benefit from additional context regarding the physiological relevance of the applied electric field parameters and a more in-depth discussion on the translational pathway from these findings to clinical application. I recommend clarifying how the structural alterations observed at the dihedral angle of GLY77 under different electric field orientations correlate with the changes in ion flux and the potential biological implications. Additionally, addressing the potential for selective targeting of cancer cells and the feasibility of practical application would greatly enhance the manuscript. The authors are encouraged to expand on these aspects to strengthen the impact and applicability of their research.

Specific comments:

Study Design and Hypothesis:

  • The study's objective to investigate the effect of THz electric fields on K+ flux through KcsA channels is commendable and relevant, given the potential for cancer therapy. However, the hypothesis would benefit from a clearer statement regarding the expected relationship between the electric field orientation and channel conduction.

Methodology:

  • The combined use of the constant electric field method (CEF) and the ion imbalance method (IIMB) for simulating transmembrane potentials is innovative. However, it would be useful to have a more detailed rationale for choosing this combined approach over either method individually.
  • The selection of the KcsA channel structural model from the Protein Data Bank and the setup of the simulation environment are well-described. It would be beneficial to discuss any limitations or assumptions in the model that could affect the results.
  • Clarity on the reasoning behind the specific amplitude and frequency of the THz field chosen for the simulations is needed. How do these parameters relate to the biological context and the potential therapeutic application?

Results:

  • The manuscript presents compelling data on how the orientation of the applied electric field affects the ion flux through the KcsA channel. However, the impact of these findings on our understanding of the KcsA channel's biophysics could be discussed in more depth.
  • The study's observation that the dihedral angle of the GLY77 residue is significantly distorted at θ=90°, affecting K+ ion permeation, is interesting. A discussion on how this specific distortion compares to physiological conditions or other perturbations would provide context.
  • While the manuscript discusses the difficulty of stabilizing the dihedral angle of GLY77, it would be beneficial to include a quantitative analysis of how these structural changes correlate with the changes in ion flux.

Discussion and Conclusions:

  • The implications for the potential therapeutic use of THz fields in cancer treatment are exciting. A more thorough discussion on the translational pathway from this basic research to clinical application would enhance the manuscript's impact.
  • The manuscript would benefit from a section discussing potential safety concerns and biological effects associated with the application of THz fields to biological tissues.
  • It is commendable that the manuscript acknowledges the limitations of simulating a single channel and attempts to extrapolate to a more realistic biological scenario. Further discussion on the challenges of scaling up from single-channel simulations to tissue-level effects would be valuable.

Suggestions for Improvement:

  • The manuscript should address the issue of selectivity in more detail. How might THz fields be targeted to cancer cells without affecting normal cells that also express KcsA channels?
  • The addition of control simulations to rule out the effects of thermal noise or other non-specific factors influenced by the THz field would strengthen the findings.
  • Consideration of the technical feasibility and design of a device capable of delivering THz fields at the required orientation and frequency in a biological setting would be a critical addition to the discussion.

Author Response

Dear Reviewer,

Thank you very much for your comments and professional advice. 

Please see the attachment.
Thank you again for your time and attention.

Best wishes,

Yize Wang

Reviewer 2 Report

Comments and Suggestions for Authors

Manuscript Review on "Effect of THz waves of different orientations on K+ permeation efficiency in the KcsA channel"

I have had the opportunity to review the manuscript entitled "Effect of THz waves of different orientations on K+ permeation efficiencies in the KcsA channel" submitted by Hongguang Wang and co-workers for consideration for publication in the MDPI journal: International Journal of Molecular Sciences. The manuscript presents a novel approach that focuses on the modulation of potassium (K+) channels using terahertz (THz) waves. It navigates through the intricate landscape of ion flux dynamics within K+ channels, emphasising their high variability and frequent alterations in various tumour types. The research not only builds upon previous findings regarding the impact of a 51.87 THz field along the Z-axis on K+ channel ion flux but also addresses the practical challenge of ensuring precise alignment between the incident electromagnetic (EM) wave and the direction of ion flow.

The practical implications of these findings are significant, with the potential to modulate K+ fluxes and induce rapid apoptosis in potassium-overloaded tumour cells. The manuscript effectively communicates the relevance of this specific THz frequency as a crucial tool in cancer therapy, offering a glimpse into a potential transformative application in practical treatment strategies.

In conclusion, this manuscript shows an innovative approach, rigorous exploration of the underlying mechanisms, and the potential it holds for advancing cancer therapy. The research paves the way for further exploration and application of terahertz waves in the treatment of cancer, presenting a compelling and promising avenue for future research in the field. The current state of the manuscript meets the necessary standards for acceptance in its present form. 

Author Response

Dear Reviewer:

Thank you for your time and attention.

We appreciate your professional comment of our article!

Best wishes,

Yize Wang

Round 2

Reviewer 1 Report

Comments and Suggestions for Authors

The authors have satisfactorily answered my questions.